# Regional frequency analysis of extreme storm surges using the extremogram approach

Marc Andreevsky[1], Yasser Hamdi[2], Samuel Griolet[3], Pietro Bernardara[4], Roberto Frau[1]

[1] Department EDF-R&D-LNHE, Chatou, France
[2] Institute for Radiological Protection and Nuclear Safety, Fontenay-Aux-Roses, France
[3] Polytech Lyon, Rochetaillée sur Saône, France
[4] EDF Energy R&D, UK Center, SW1E5JL, UK

*Correspondence to:* Marc Andreevsky (marc.andreevsky@edf.fr)

**Abstract.** To withstand coastal flooding, protection of coastal facilities and structures must be designed with the most accurate estimate of extreme storm surge return levels (SSRLs). However, because of the paucity of data, local statistical analyses often lead to poor frequency estimations. The regional frequency analysis (RFA) reduces the uncertainties associated with these estimations, by extending the dataset from local (only available data at the target site) to regional (data at all the neighbouring sites including the target site) and by assuming, at the scale of a region, a similar extremal behaviour. In this work, the empirical spatial extremogram (ESE) approach is used. It is a graph representing all the coefficients of extremal dependence between a given target site and all the other sites in the whole region. It allows quantifying the pairwise closeness between sites based on the extremal dependence. The ESE approach, which should help being more confident about the physical homogeneity of the region of interest, is applied on a database of extreme skew storm surges (SSSs) and used to perform a RFA.

Keywords: Regional Frequency Analysis, Homogeneous region, Target site, Spatial-extremogram, Storm surges.

## 1 Introduction

To resist flooding hazard in coastal areas, considering the most accurate frequency estimates of extreme storm surges return levels (SSRLs) (1000-year return level, for instance) with an appropriate confidence level becomes a major operational concern when designing protections. The *T*-year return level can be defined as a high quantile for which the probability that an extreme value (the annual maximum, for instance) exceeds this quantile is $\frac{1}{T}$. When performing a local frequency analysis, the size of the sample is often too low to obtain accurate estimations of high SSRLs. For example, storm surge at-site records calculated from tidal gauge signals are usually shorter than 30 years. The associated uncertainties can be reduced by a Regional Frequency Analysis (RFA) which attempts to exploit the similarities between sites and deals with the estimation of hydrological characteristics at sites where little or even no data is available. The RFA introduced by Dalrymple (1960) is based on the index flood method under the assumption that within a homogeneous region, extremes (normalized by a local index) are drawn from a common regional distribution. The grouping of sites into homogeneous regions defines the way to exploit regional information and, then, can have a significant impact on final results. Numerous papers have tried to tackle this issue in hydrology by studying the explanatory variables representative of the phenomena of interest (e.g. GREHYS 1996a, b, Hosking and Wallis, 1993, 1997, Chebana and Ouarda, 2008; Das

and Cunnane, 2011,). The index flood concept with the approach developed by Hosking and Wallis (1993, 1997) was extensively applied in the literature especially in hydrology to characterise river flood hazards (e.g. Hosking and Wallis, 1997). In order to characterize coastal hazards and to estimate extreme skew storm surges (SSSs), a RFA has been recently applied (e.g., Bardet et al., 2011; Bernardara et al., 2011). For convenience, we would like to recall here the definition of a skew surge: it is the difference between the maximum observed water level and the maximum predicted tidal level regardless of their timing during the tidal cycle (a tidal cycle contains one skew surge). Other authors recommended the use of meteorological information to delineate homogeneous regions and to carry out RFA of rainfall (e.g. Gabriele and Chiaravalloti, 2013). The criteria of merging sites in a homogeneous region were mainly based on statistical arguments, thereby excluding physical considerations. For example, by using data from 18 sites located on the west French coasts, Bernardara et al. (2011) used a statistical test of regional homogeneity to form the whole area of interest.

More recently, Weiss (2014c) introduced a physically based method to delineate homogeneous regions in order to perform a RFA of the extreme SSSs. This method depends on the storm footprints identified through a declustering algorithm using a storm propagation probabilistic criterion. However, if a target site is very close to the limit of the region, the information at the site located on the other side of the region of interest can be wrongly excluded, even though both sites likely offer similar information and have likely similar asymptotic properties. This problem is also known as the "border effect". For instance, in the regions of interest obtained by Weiss (2014c), the two French sites, Boulogne-Sur-Mer and Calais are located in two different regions, while they are geographically close with a distance of about 30 km. Indeed, despite the fact that both sites face different seas (North Sea and English Channel), they have the same climate (according to the climate comparator proposed by Meteo-France: http://www.meteofrance.com/climat/comparateur). One can also notice that there are several other areas in which sites with similar statistical and physical behavior are located on both sides of a region border. Moreover, very distant sites can be gathered in the same region, which could involve traces of heterogeneity, even in a region considered statistically homogeneous. Acreman and Wiltshire (1987) have suggested that the sites located near the border between 2 regions could be considered partly owned by each region. However, Burn (1990) suggests that there is no need to define boundaries between regions and a particular region can be defined for each site (which consists of sites similar to the site of interest in terms of extremes).

To address this limitation (the border-effect problem) and to form a physically homogeneous region centered on a target site, we take up in the present paper an approach, which was proposed by Hamdi et al. (2016), using the empirical spatial extremogram (ESE) in which the extremal dependence between two observation series becomes a measure of the neighbourhood between the two associated sites. A pairwise measure between sites based on the spatial extremal coefficients was defined to carry out a RFA applied on extreme SSSs. The composition of regions built here is based on the similarity of sites' attributes. The higher the value of the spatial extremal coefficient between the target site and another site is, the greater the dependency of extreme SSSs, therefore indicating that storms impacting the target site tend to also impact the other site which can be included in the region of the target site. Indeed, in a region of interest, the process generating storms and impacting the target site will tend to impact the other sites in the region as well and vice versa. It is with this in mind that the processes generating storms in a region are considered physically homogeneous. And then, it is assumed that sites, with sufficiently high spatial extremal coefficients (with the target site), may be included in the same

region of influence of the target site (the physically homogeneous region). The region may also be considered as a typical storm footprint in the neighbourhood of the target site. Obviously, the dependence between sites must be taken into account in the statistical analysis. Once a physically homogeneous region (centered on the target site) is formed the statistical homogeneity is then checked and the regional frequency estimation (and, in particular, the dependence model and the way to calculate the effective duration of observations) is subsequently performed.

Our objective in this work is to conduct a RFA using the ESE. The ESE approach should enable us to get rid of the border effect and the distance impact, and also to be more confident about the physically homogenous aspect of the region. The paper is organized as follows. A description of the methods is presented in Sect. 2. The case study with SSSs data in the whole region is also presented in Sect.2. In the section 3, the ESE will be applied to some target sites. The results of the analysis are further discussed in Sect. 4, before the conclusion and perspectives in section 5.

## 2 Materials and methods

### 2.1 The Empirical Spatial Extremogram (ESE)

The objective of the present section is to use an approach based on the spatial extremogram to form a homogenous region to be used in a RFA of extreme SSSs. The region of interest which is expected to be centered on a given target site must be physically and statistically homogeneous. The use of the spatial extremogram technique in index flood based RFA is the main contribution of the present paper. It is expected that the spatial extremogram based procedure leads to regions of interest with no border effect and with less residual heterogeneity.

Let $X$ and $Y$ be the random explanatory variables of the SSSs at sites $S_1$ (the target site) and any other site $S_2$, respectively. Let $q_1$ and $q_2$ be the two samples extreme quantiles (thresholds above which SSSs are considered as extremes) for sites $S_1$ and $S_2$, respectively. Site $S_2$ is inside the physically homogeneous region centered on the target site $S_1$ if, at least, a certain part of extreme SSSs from each site are likely to be simultaneously induced by the same storms. This means that there is an extremal dependency between both sites. In the spatial and pairwise dependence description the temporal dimension should be included, because storm conditions can last several days. The ESE compute is then based on the pairwise extremal dependences (between any site and the target site) and it can be defined by the spatial extremal coefficient $\rho(X,Y)$, as proposed by Hamdi $et\ al.$ (2016) and expressed as:

$$\rho: R \times R \rightarrow [0,1] \qquad \rho(X,Y) = \lim_{n \to \infty}\left(P\left[X \geq q_1 \mid Y \geq q_2\right]\right) \tag{1}$$

The empirical and spatial extremal coefficient is the natural estimator $\rho$ of $\rho$ and it is defined by:

$$\hat{\rho}(X,Y) = \frac{\sum_{t=1}^{D} I\{X(t) > q_1\ \&\ Y(t) > q_2\}}{\sum_{t=1}^{N} I\{X(t) > q_1\}} \tag{2}$$

Where $I\{f\}$ is equal to 1 if $f$ is true, and to 0 otherwise, $N$ is the number of extreme events $X(t) > q_1$ at the target site (events that exceeded the sample extreme quantile $q_1$) and $D$ is the number of overlapped extreme twins $\{X(t) > q_1\ \&\ Y(t) > q_2\}$, extreme events generated by the same storm that occurred at sites $S_1$ and $S_2$. It was considered that two observations were generated by the same storm if the time difference between their two moments of occurrence was less,

in absolute value, than 6 hours (other durations were tested, 12h and 24h but the results were similar). It is noteworthy that D must be large enough, in order for ESE compute to be reliable. Indeed, the larger it is, the more significant the probability of extremal dependence. The fact that $\hat{\rho}(X,Y)$ ranges between 0 and 1 indicates the existence of a certain extremal dependency between $X$ and $Y$. The higher the empirical extremal coefficient $\rho$, the stronger is the dependency between both sites. Indeed, we obtain an extremal coefficient equal to 1 in case of perfect dependency between sites (theoretically, this can only happen when computing the extremal dependency between a site and itself). But more interestingly, $\rho$ is relatively high when a storm is observed at site $S_1$ then, relatively often; this storm is also observed at site $S_2$. However, the extremal coefficient tends towards 0 when the site of interest is far enough from the target site. In other words, the observations at sites, generally widely separated geographically, can be considered as asymptotically independent. An illustrative example on how the extremal dependency coefficient is empirically computed is presented in Fig. 1. As it can be shown in this example, among the 7 extreme twins ( $N = 7$ ), only 4 overlapped with the target site extreme storm surges ( $D = 4$ ). The extremal coefficient is then equal to 0.57. Obviously, this is just an illustration and as it was mentioned earlier in this section, the number of extreme twins $N$ and overlapped ones $D$ must be large enough.

The scheme for obtaining the extremal dependence between a target site and all the other sites has to be applied differently for each case study. The first question one can ask is from what value of the extremal dependency coefficient two sites begin to be considered as neighbors? This brings up some other questions: when the extremal correlation begins to be statistically significant? Or from which value of $\rho_0$ we begin to have a positive association between two sites (when they experience a significant simultaneous rise in water during a storm)? Objective answers to these questions cannot in any case suggest an exact value or a statistic under or above which things are different but there is an area in the space of the parameter or a range of values that can be explored and depending on the sensitivity to this coefficient conclusions can be drawn.

Let $\rho_0$ be the threshold above which the probability of extremal dependence between $X$ and $Y$ is high enough to consider $S_1$ and $S_2$ inside the same physically homogeneous region. First and foremost, there is a trade-off associated with the selection of this threshold: a large value of $\rho_0$ can result in a too small homogeneous region, but the opposite is likely to cause a residual probability (nothing other than a noise) which interferes with the construction of the region. The threshold $\rho_0$ can be estimated by analysing the pairwise extremal dependence probability (between all sites and the target site). This procedure allows for the evaluation of the maximum residual noise order of magnitude. In addition, the determination of "an optimal" $\rho_0$ has to allow for the data merging of the neighbouring sites of the target site (which have an extreme dependence on it), and put aside the "false" neighbours (sites having a residual extremal dependence probability with the target site considered as a noise). The empirical quantiles $q_1$ and $q_2$ are set in order to extract in $X$ and $Y$ series only extreme events per year, which allows for the computation of the empirical spatial extremal coefficient from the biggest storms of each year. We know intuitively that the threshold $\rho_0$ may vary depending on the climate of the region and other physical and physiographic factors such as the bathymetry and the presence or not of a tidal estuary. Finally, the target site neighbourhood contains all sites, which have a probability of extremal dependence greater than $\rho_0$. From a physical point of view, this means that when a storm affects a given target site, it will likely impact (not systematically) only sites enclosed

in the region of interest and vice versa. The neighbourhood of the target site can also be considered as the region of influence around the target site as introduced by Burn (1990).

## 2.2 Independent storms extraction and construction of the regional frequency model

Once the homogeneous region of interest centered on the target site is obtained, the procedure begins by constructing a regional sample of independent storms. A storm is defined as a physical event that induces extreme SSSs (i.e. exceeds the

extreme quantile $q_p$ ) in at least one site in the region of interest. To extract independent extreme SSSs, only the maximum value is kept.

The RFA uses the flood index principal which stipulates that within a statistically homogeneous region, extreme events normalized with a local index are drawn from a common regional distribution. It is assumed that the distribution of these extreme normalized SSSs converges to a Generalized Pareto Distribution (GPD) and the number of exceedances converges

to a Poisson distribution. The annual SSS quantile was used as a local index to normalize at-site samples. A further noteworthy statistical setting of the developed RFA is that it uses a relatively high threshold, allowing the selection of extreme storms corresponding to an annual rate $\lambda$ of extreme SSSs equal to 1. To meet and satisfy the statistical homogeneity requirement of the region of interest, two L-moments based tests introduced by Hosking and Wallis (1997) are used: (i) the heterogeneity indicator $H$ which is a measure of whether the dispersion between sites is similar to a value

that would be expected in a statistically homogeneous region; (ii) the discordancy criterion $D_c$ to ensure that any site is not significantly different, in terms of L-moments, from the other sites. Hosking and Wallis suggest that a site is discordant if $D_c < 3$. In addition, a region is considered "acceptably homogeneous" if $H < 1$, "possibly heterogeneous" if $1 \leq H < 2$, and "definitely heterogeneous" if $H \geq 2$. These tests are performed for each region of interest. A site is generally eliminated from the inference if it is found to be discordant. To make the best possible use of the expertise (to be more

conservative, for instance), the results are compared with and without the inclusion of discordant sites.

## 2.3 Regional effective duration of observations

In a regional context, the effective duration of observations $D_{eff}$ of the regional sample depends on those of local samples $d_i (i = 1,...N)$, where $N$ is the number of sites in the region of interest. Indeed, $D_{eff}$ must be filtered of any intersite or any spatial dependencies and correlations. If at-site samples in a given region are completely independent, pooling data

from these sites leads to a regional effective duration of observations equal to $\sum d_i$. This is obviously not the case herein. In those cases where the intersite dependence is perfect, $D_{eff}$ can be formulated as the average of durations of observations $d_i$. Weiss *et al*. (2014b) have developed a spatial function $\varphi$ that reflects the intersite dependence in a region of interest. Weiss *et al*. (2014b) have shown that $\varphi = \lambda_r / \lambda$, where $\lambda_r$ is the average annual number of storms in the region and $\lambda$ is the average number of storms per year at each site. $D_{eff}$ is expressed as a weighted sum in the following form:

$$D_{eff} = \frac{\varphi}{N} \sum_{1}^{N} d_i = \frac{n_r}{\lambda} \tag{3}$$

Where $n_r$ is the number of regional SSSs. The function $\varphi$ takes a value between 1 and $N$ according to the level of dependence in the region:

- $\varphi = 1$ if the region is perfectly regional-dependent. The effective duration of observations $D_{eff}$ takes then the form of the average of durations: $D_{eff} = \frac{1}{N}\sum d_i$ .
- $\varphi = N$ if the region is regional-independent. This leads to: $D_{eff} = \sum d_i$

## 2.4 Fitting the regional SSSs and at target-site frequency estimations

The regional frequency model used herein is based on the extreme value theory. As mentioned in the section 2.2, the peaks-over-threshold (POT) approach (*e.g.*, Pickands, 1975), in which the excesses are analyzed with the GPD, is used in the RFA using a threshold equal to 1 and taking into account the seasonality. Seasonal effects, considered for other variables in the literature (*e.g.* Morton, 1997, Méndez, 2007), can be modelled through a sinusoid. The regional distribution becomes a discrete mixture of GPD/sinusoid with a seasonal-varying scale parameter $\xi$ ($\xi$ varies periodically and smoothly across the seasons) (Weiss, 2014c). The probability distributions are constructed as follows:

Let's denote $v$ the regional random variable. A GPD is fitted to the regional sample taking into account the four seasons in the following way:

$$\forall v \geq 1, F_r(v) = \sum_{c=1}^{4} p_{r,c} F_{r,c}(v) \tag{4}$$

$p_{r,c}$ is the frequency of occurrence of season c in the regional sample (empirically estimated as the observed proportion of storms that occurred during season c in the region). We also have:

$$F_{r,c} \sim GPD(1, \gamma_{r,c}, k_r) \tag{5}$$

and

$$log(\gamma_{r,c}) = \gamma_r^0 + \gamma_r^1 \cos(\frac{2\pi}{4}c) + \gamma_r^2 \sin(\frac{2\pi}{4}c) \tag{6}$$

Where $\gamma_r^0$, $\gamma_r^1$ and $\gamma_r^2$ are *3* parameters to be estimated. Table 3 presents the 8 models derived from the possible values of $\gamma_r^0$, $\gamma_r^1$, $\gamma_r^2$ and $k_r$. Besides the visual inspection of the fitting curve, the Akaike Information Criterion (AIC) is used to select the more adequate distribution. Finally, the frequency estimation can be easily reverted to estimate the at-target site SSRLs by multiplying the regional ones by the local index (*i.e.* the annual SSS quantile).

## 2.5 The case study

SSS datasets are obtained from the temporal series of hourly observations and predicted tide levels (the astronomical tide), collected at a total of 67 sites located on the the Spanish, French (Atlantic and English Channel) and British coasts (see Fig. 2). The French tide gauges are managed by the French Oceanographic Service (SHOM - Service Hydrographique et Océanographique de la Marine) while Spanish and British ones are managed by IEO (Instituto Español de Oceanografía, Spain) and the British Oceanographic Data Centre (BODC), respectively.

For convenience, the same observation periods as those used by Weiss (2014c) have been used in the present study. They range from 1846 for Brest (in France) to 2011 (for almost all the sites) and they show an average effective duration of observations of 31 years. In most cases, local series are characterized by the presence of many gaps. It is to be noticed that

the sea levels must be corrected from a possible eustatism (a general variation in mean sea level) not to induce bias in the calculation of the surges: a correction is done if annual sea levels (calculated following the Permanent Service for Mean Sea Level recommendations) show significant trends. It is also noteworthy that the impact of climate change on the estimated return levels and associated uncertainties is not covered by this paper. The use of projected sea level rise could however be the object of another paper.

Furthermore, in connection with the choice of the variable of interest, the focus is restricted to SSS series because in regions with strong tidal influence, the coastal flooding hazard is most noticeable around the time of high tide. Indeed, the SSS is a fundamental input for many statistical investigations of coastal hazards. It is defined as the difference between the maximum observed sea level and the one predicted around the time of high tide. Thus, the resulting SSSs series have a temporal resolution of approximately 12.4 hours. The reader is referred to Bernardara et al. (2011) for a more detailed introduction on skew surges. The developed RFA is performed at many target sites along the French (Atlantic and English Channel) coast. One of the most important features of these target sites is the fact that the region in which they are located has experienced significant storms during the last few decades (1953, 1987, Lothar and Martin in 1999 and the Xynthia in 2010). Fig. 2 displays the geographic location of the whole region. As depicted in the left-hand panel of the figure, three target sites (red empty circles) are selected to perform the developed methodology and estimate the 1000-year return level.

## 3. Results

All the simulations are carried out within the R environment (open-source software for statistical computing: http://www.r-project.org/). The main results of the developed RFA with all the diagnostics are presented in terms of tables and figures (probability plots), wherein the main focus is set on the storm surge quantile corresponding to the return period T = 1000 years and the width of the 70% confidence interval. Prior to the RFA, the results of the homogenous region delineation are presented first.

### 3.1 Formation of regions of interest centered on target sites

Two types of thresholds are used in the calculation of the empirical spatial extremogram. The first threshold sets the extreme quantiles to extract extreme SSSs and the second one (the neighbourhood threshold) sets the extremal coefficient above which sites are considered neighbours. Since too high thresholds result in introducing a high variance and a too low one introduces a bias in the results, there is a trade-off to be made between variance and bias. Indeed, the asymptotic properties of the marginal SSSs can be violated when a too low extreme quantile $q_p$ ( $p < 70\%$ , for instance) is used to compute the extremal dependence coefficient of a given site. The use of a too high threshold ( $p > 99\%$ , for instance) can significantly reduce the number of the pairs of simultaneous extreme events to be used to compute the extremal coefficient. It is to be noticed that, even if the $q_p$ threshold is adequately selected, a too high neighbourhood threshold ( $\rho_0 > 0.7$ , for instance) will limit the number of neighbouring sites and decrease the size of the region of interest while a too low threshold will likely cause a residual probability (which is nothing other than a noise) and will erroneously increase this region. Values of $q_X$ and $q_Y$ are tested in order to select 4 and then 6 storms a year, which finally give information from the empirical spatial extremogram that lead to similar regions. Therefore, the extremal quantiles are set to select only 4 storms a year. This value allows for the computation of the empirical spatial extremogram from the biggest storm of each year.

Moreover, the lag time $h$, has to be large enough to allow a storm which occurs at one site to propagate eventually to the other site. Spatial extremograms performed with lag times greater than 24 hours show little difference compared to those obtained with lag times equal to zero and to the duration between two SSSs (about ±12.4 hours). The latest two lag times are then used and the greatest value among the corresponding extremal coefficients is kept. Finally, the neighbourhood threshold is set to 0.3. This value allows for the elimination of any sites associated with a value of the spatial extremal coefficient that look like a residual noise.

*Calais (station number 24):*

One of the most important features of the Calais site is the fact that it is located close to a border of one of the regions found by Weiss (2014c). In addition, the region in which this site is located has experienced significant storms during the last two decades (Martin in 1999 and the Xynthia in 2010). Fig. 3 displays the geographic location of five homogeneous regions according to Weiss, (2014c). The scheme for obtaining the pairwise extremal dependence coefficients between Calais as a target site and all the other sites is applied herein. From the ESE depicted in the top panel of Fig. 4, a geographically coherent region of interest corresponding to the neighbourhood threshold $(\rho_0 = 0.3)$ is obtained and illustrated in the left panel of Fig. 5. As it can be seen in Fig. 4, the pairwise probabilities of the extremal dependence between Calais and all the sites in the whole region are presented on the vertical axis. The sites of the whole region, presented on the x-axis, are sorted in an ascending order based on the geographical distance to the target site. The physically homogeneous group of sites with extremal dependence probabilities greater than $\rho_0$ (the red lines on Fig. 4) are considered as potential neighbours of the target site (Calais) and are thus part of the region of interest (of Calais). Pairwise simultaneous length of record (at the target site and any site of the whole region) appears in brackets next to the name of each site in Fig. 4. This duration is an important setting because it is the number of years on which the spatial extremal coefficients are calculated. For instance, the time during which Calais and Dunkerque (station number 25) operated simultaneously is equal to 26 years. It is noteworthy that the probability of the extremal dependence may not be relevant if this time is too small, and whether it is appropriate to consider the site in the inference will be assessed on a case-by-case basis. As shown in the left panel of Fig. 5, the target site Calais is no longer located at the border of the region. Indeed, the physically homogeneous region of interest around Calais is slightly smaller than the one obtained by Weiss (2014c) but more centered on Calais. Further noteworthy features of the ESE is that it provides regions smaller than those obtained by Weiss (2014c) and then more physically homogeneous.

*Brest (station number 15):*

The ESE for the whole region with Brest as a target site is shown in Fig. 4 (to the middle) and the associated region of interest is depicted in the middle panel of Fig. 5. As shown in this figure, the region of interest around Brest is larger than the one centered on Calais with many sites for which the extremal dependence coefficients are at the limit of the neighbouring threshold $\rho_0$. It is definitely so with sites located in the Bristol Channel on the British coast. A sensitivity study is conducted with and without these sites and it is concluded that these sites should remain in the region of interest. Indeed, their absence led to less adequate fitting. As mentioned earlier and as shown in the middle panel of Fig. 5, the homogeneous region centered on Brest is smaller than the one obtained by Weiss (2014c) but it is nevertheless better centered on the target site Brest.

*La Rochelle (station number 8):*

One of the most important features of the La Rochelle site is the fact that the region in which this site is located has recently experienced the Xynthia storm (2010). It has been the subject of many studies after this storm (*e.g.* Hamdi *et al*., 2015). Fig. 4 (to the bottom) shows the ESE (with La Rochelle as a target site) and in the right panel of Fig. 5 is depicted the homogeneous region of interest centered on La Rochelle. As concluded with the two first target sites Calais and Brest, it can be seen in the right panel of Fig. 5 that the region of interest is also smaller than that obtained by Weiss (2014c) but better centered on the La Rochelle site. The time during which La Rochelle and both Saint-Malo (station number 18) and Saint-Servan (station number 17) sites operated simultaneously is relatively small (14 years for Saint-Malo and 2 years for Saint-Servan). The Extremal dependency coefficients for these two sites are equal to *0.29* and *0.4,* respectively. The question of whether to consider Saint-Servan and Saint-Malo inside the region or not, has been raised. Since both sites are very close to each other (with a distance of less than *2* km), it seems logical to either add them both to the region, or withdraw them both from the region. We finally decided to integrate them into the region of interest because the site Jersey is part of the region centered on La Rochelle with a dependency extremal probability equal to the neighbouring threshold ($\rho_0 = 0.3$) and a common period of only 14 years. The site Jersey, abbreviated "JER" in the ESE plots with the station number 19, is geographically very close to Saint-Malo and Saint-Servan.

Once the physically homogeneous regions are formed, the statistical homogeneity must be verified. As mentioned earlier in this paper, the L-moments based homogeneity tests (heterogeneity measure and discrepancy) are used. The heterogeneity measure *H* is equal to -0,13 for Calais, 0.99 for Brest and 1.13 for La Rochelle. The only case where there has been a discordant site is when La Rochelle is the target site. Indeed, with a $D_c$ of 3.65, the Eyrac site (station number 5) has been identified as discordant. This site is located in the center of the region and this discrepancy could be explained by the specific sea conditions in the Arcachon basin. It is noteworthy that when Eyrac is removed from the region, the heterogeneity measure *H* becomes equal to 0.53. So a new region without Eyrac considered statistically homogeneous is used.

### 3.2 The regional and local frequency estimations

As mentioned earlier in this paper, a regional pooling method to estimate the regional distribution for each homogeneous region is used. Indeed, a storm that can impact several sites (thus generating intersite dependence) during a single storm is considered only once in the regional sample. The distribution of the maximum regional SSSs ($M_s$) is assumed to be identical to the regional distribution. In order to verify the validity of this assumption, a Kolmogorov-Smirnov test (*H<sub>o</sub>*: the regional observations and $M_s$ sample follow the same distribution) is performed. The null hypothesis *H<sub>o</sub>* is satisfied for the three regions centered on Calais, Brest, and La Rochelle at a risk level of 5%. Consequently, the regional distribution can be estimated from each regional sample $M_s$. The way the delineation of the homogeneous regions is performed implies that the regional sample is characterized by a strong spatial dependence which impacts, in a first step, the regional effective duration of observations *D<sub>eff</sub>* (that will be even lower if this dependence is high). The effective duration of observations *D<sub>eff</sub>* is calculated using Eq. 3 and results are compared with those obtained by Weiss (2014c). These results are summarized

in Table 1. It is important to remember here that in the study of Weiss (2014c), Calais has been a part of region 2 while Brest and La Rochelle are located in region 1. As shown in Table 1, the regional effective duration of observations $D_{eff}$ associated with the region of Calais is the same in both studies. However, it is smaller in the present work for the region of Brest and La Rochelle. In any case, however, regional effective durations are obviously higher than local ones. Furthermore, a Student test is performed to check if the regional sample is stationary in intensity (two subsamples have equal means) or not. It is concluded that all the samples are stationary in intensity.

A GPD distribution taking into account the seasonality is then fitted to the regional sample. The distribution parameters are estimated with the penalized maximum likelihood method (Coles and Dixon, 1999). The most adequate distributions are obtained with the AIC criterion. The $Exp_{sin}$ distribution is selected for Calais while it is concluded that the distribution of the extreme regional SSSs for Brest and La Rochelle converge to a $Gpd_{cos\ sin}$. The same frequency models are selected by Weiss (2014c) for regions 2 (including Calais) and 1 (including Brest and La Rochelle), respectively. The reader is referred to Weiss (2014c) to learn more about the mixture GPD-sinusoid distributions with a seasonal-varying scale parameter. The Fitting curves for Calais, Brest and La Rochelle, which are shown in Fig. 6, looks good. Indeed, most of the points in the body of the distribution are inside the confidence intervals. Those elements are also relevant to accept the credibility of the distributions used for the fitting.

In the next RFA step, local quantiles are estimated by multiplying the regional ones by the local indices. The results are summarized in Table 2 presenting a comparison of the 1000-year return levels with associated confidence intervals

It is worth concluding that better centering the region of interest on the Calais site did not significantly change quantiles (a decrease of only 7 cm) but it rather narrowed the associated confidence interval of about 12 cm. This outcome refers only to the region of interest around Calais and it is different for the region centered on Brest and La Rochelle sites. However, the quantiles and associated confidence intervals are overall roughly the same, but the method presented herein better answers the uncertainties linked to the border effect issue, notably through the ESE tool.

## 4. Discussion

One of the most important features of the ESE based approach used in this paper to form a physically homogenous region centered on a target site is the fact that it avoids the problem of the so called "border effect". Moreover, and in contrast to that, introduced by Weiss (2014c), the extremogram tool seems to prevent too distant sites to belong to the same homogeneous region. This reduces physical and statistical heterogeneity that could be generated by pairwise sites quite far apart. Consequently, the spatial extremogram approach offers the key advantage to lead to a certain geographical consistency. Despite the fact that the 1000-year return level and associated confidence interval obtained in this work are close to those obtained by Weiss (2014c), the spatial extremogram method improves the physical homogeneity of the regions of interest and can decrease the effective duration of observations. Nevertheless, findings for the sites of Calais (which is no longer close to the border of a region) and Dunkerque seem to be particularly interesting for us because they are no longer close to the border of a region and since they can be representative sites for the Gravelines nuclear power plant in France. Furthermore, physical homogeneity may have an impact on the statistical one. Indeed, by using the L-moments based criteria (Hosking and Wallis, 1997), it is concluded that unlike the regions 1 and 2 in Weiss (2014c) (which

are considered as possibly statistically homogeneous), all the regions built herein are statistically homogeneous, which is also a progress.

The ESE based approach can, nevertheless, be limited by the size of the common pairwise time period (during which data is present in both sites). Indeed, when the tide gauge at two different sites is often not operational over different periods of time, the common time period between these two sites used to calculate the spatial extremal coefficient may be short. Thus,

sometimes a spatial extremogram can be considered as not relevant. Therefore this shortcoming must be taken into account during the formation of the regions of interest, for instance by possibly removing the site involved. It will therefore be interesting to analyze the uncertainties related to the ESE approach in order to have more reliability on the estimates of the extremal dependence between sites.

## 5. Conclusions

This study aims to perform regional frequency estimations of SSSs as an alternative to the local frequency analysis. Several ideas and approaches have been proposed in the literature to tackle the issue of the delineation of homogeneous regions which is a main step in a RFA. The present work provides a detailed reasoning for the need to use a more robust and reliable method which allows for the delineation of one homogeneous region centered on a target site, and utilizes a method based on the calculation of pairwise extremal dependence coefficients (the empirical spatial extremogram) introduced by

Hamdi *et al*. (2016) and compares the results with those obtained by Weiss *et al*. (2014c). The regional sample of the independent maximum normalized SSSs is then constructed from the series of the concordant sites in the region of interest. A regional effective duration of observations reflecting the intersite dependence of this sample is subsequently calculated and used in the regional frequency estimations.

Another consideration in this paper is applying and illustrating the ESE approach on a whole region containing sites located

on the Spanish, French (Atlantic and English Channel) and British coasts with three target sites in France (Calais, Brest and La Rochelle). A regional mixture of GPD/sinusoid distribution with a seasonal-varying scale parameter and confidence intervals are examined. Overall, the results suggest that the regional analysis can be helpful in making a more appropriate assessment of the risk associated with the coastal flooding hazard. The application demonstrates also that the RLs and associated confidence intervals estimates for Calais, Brest and La Rochelle target sites are close to those obtained in a

previous work (Weiss, 2014c).

An in-depth study using more physical data and criteria in addition to the ESE (such as the atmospheric pressure or the wind speed and direction) could help to form regions more physically homogeneous. The concept of the ESE should find additional applications for the assessment of risk associated with other hazards in other climate and geoscience fields (*e.g*. extreme temperature and heat wave hazards). Associating confidence intervals with the spatial extremal coefficients could

also be interesting. Another possible future endeavor is to perform a RFA using a regional sample containing all the regional SSSs (not only the maximum per storm and without considering the intersite dependence).

**Conflict of interest:** We claim that all authors agree with the submission of this manuscript. We also claim that this manuscript has not been published before and is not concurrently being considered for publication elsewhere. This

manuscript does not violate any copyright or other personal proprietary right of any person or entity and it contains no
statements that are unlawful in any way.

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

**Table 1.** Comparison of total and effective durations of observations (years) used in the present study with those used by Weiss (2014c). The effective duration of observations takes into account the intersite-dependence. A total duration of observations is the sum of all at-site durations in the region of interest (without intersite dependence).

| | The present work | | | Weiss (2014c) | |
|---|---|---|---|---|---|
| | Calais | Brest | La Rochelle | Region 1(Calais) | Region 2 (Brest & La Rochelle) |
| $D_{eff}$ | 151 | 348 | 348 | 517 | 151 |
| Total duration | 375 | 783 | 605 | 1011 | 443 |

**Table 2.** Comparison of the 1000-year return levels with the 70% of the confidence interval width (70%CI) in brackets obtained herein with those of Weiss (2014c)

| | 1000 year-RLs (70%CI) (m) | | |
| --- | --- | --- | --- |
| | Calais | Brest | La Rochelle |
| Results from present study | 1.55 (0.18) | 1.68 (0.19) | 1.68 (0.17) |
| Results from Weiss (2014c) | 1.62 (0.30) | 1.56 (0.14) | 1.67 (0.15) |

**Table 3.** Possible models for the fitting of the regional samples.

| Model Names | Parameters values |
| --- | --- |
| Exp | $\gamma_r^1 = \gamma_r^2 = k_r = 0, \gamma_r^0 \epsilon R$ |
| Expcos | $\gamma_r^2 = k_r = 0, (\gamma_r^0, \gamma_r^1) \epsilon R^2$ |
| Expsin | $\gamma_r^1 = k_r = 0, \ (\gamma_r^0, \gamma_r^2) \epsilon R^2$ |
| Expcos sin | $k_r = 0, (\gamma_r^0, \gamma_r^1, \gamma_r^2) \epsilon R^3$ |
| GPD | $\gamma_r^1 = \gamma_r^2 = 0, (\gamma_r^0, k_r) \epsilon R^2$ |
| GPDcos | $\gamma_r^2 = 0, (\gamma_r^0, \gamma_r^1, k_r) \epsilon R^3$ |
| Gpgsin | $\gamma_r^1 = 0, (\gamma_r^0, \gamma_r^2, k_r) \epsilon R^3$ |
| GPDcos sin | $(\gamma_r^0, \gamma_r^1, \gamma_r^2, k_r) \epsilon R^4$ |

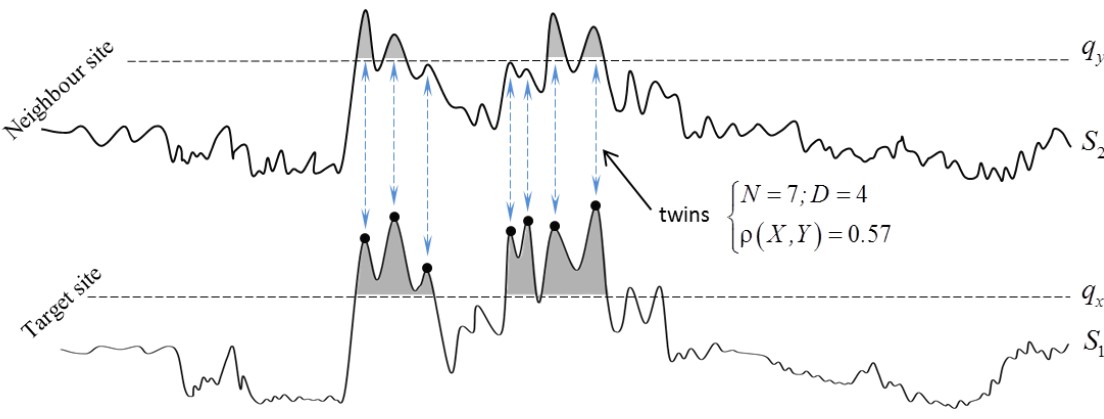

**Fig. 1.** An illustrative example on how the extremal dependency coefficient is empirically computed. Among the 7 extreme twins ( $N = 7$ ), only 4 overlapped with the target site extreme storm surges ( $D = 4$ ). The extremal coefficient is then equal to 0.57.

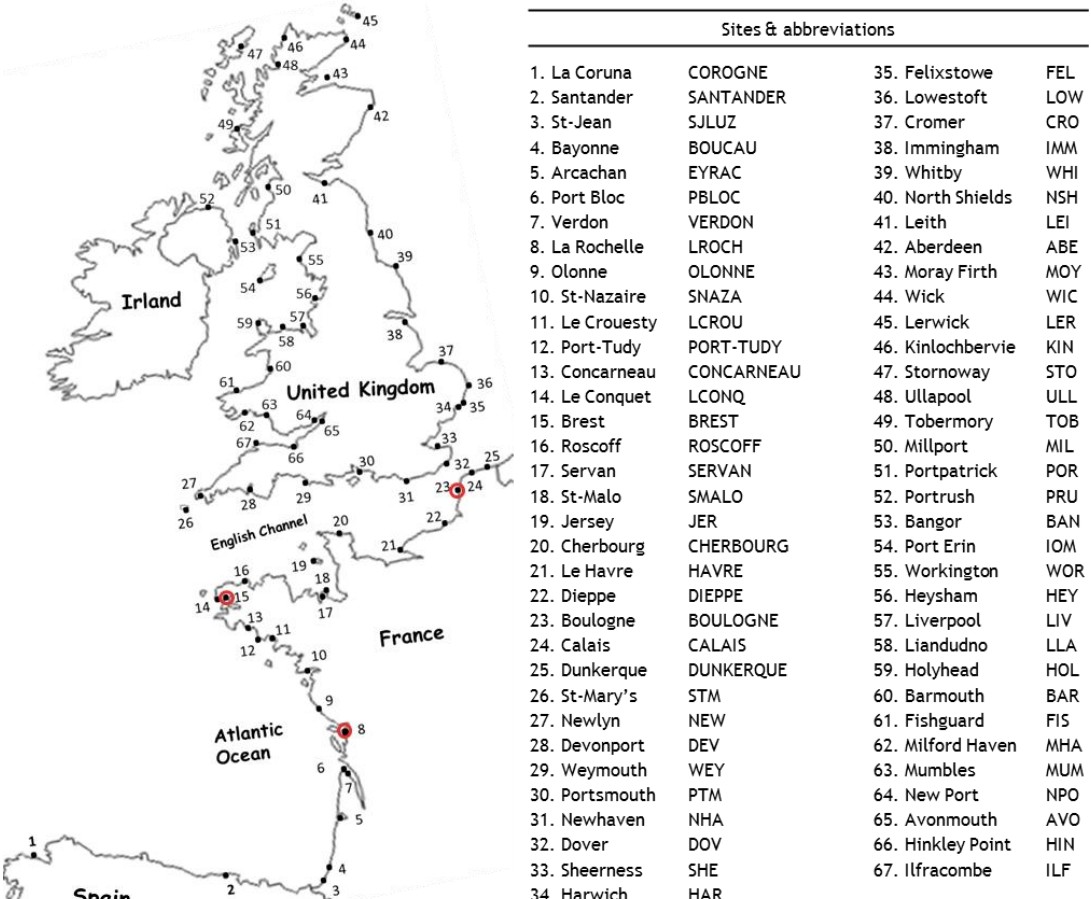

| Sites & abbreviations | | | | |
|---|---|---|---|---|
| 1. La Coruna | COROGNE | | 35. Felixstowe | FEL |
| 2. Santander | SANTANDER | | 36. Lowestoft | LOW |
| 3. St-Jean | SJLUZ | | 37. Cromer | CRO |
| 4. Bayonne | BOUCAU | | 38. Immingham | IMM |
| 5. Arcachan | EYRAC | | 39. Whitby | WHI |
| 6. Port Bloc | PBLOC | | 40. North Shields | NSH |
| 7. Verdon | VERDON | | 41. Leith | LEI |
| 8. La Rochelle | LROCH | | 42. Aberdeen | ABE |
| 9. Olonne | OLONNE | | 43. Moray Firth | MOY |
| 10. St-Nazaire | SNAZA | | 44. Wick | WIC |
| 11. Le Crouesty | LCROU | | 45. Lerwick | LER |
| 12. Port-Tudy | PORT-TUDY | | 46. Kinlochbervie | KIN |
| 13. Concarneau | CONCARNEAU | | 47. Stornoway | STO |
| 14. Le Conquet | LCONQ | | 48. Ullapool | ULL |
| 15. Brest | BREST | | 49. Tobermory | TOB |
| 16. Roscoff | ROSCOFF | | 50. Millport | MIL |
| 17. Servan | SERVAN | | 51. Portpatrick | POR |
| 18. St-Malo | SMALO | | 52. Portrush | PRU |
| 19. Jersey | JER | | 53. Bangor | BAN |
| 20. Cherbourg | CHERBOURG | | 54. Port Erin | IOM |
| 21. Le Havre | HAVRE | | 55. Workington | WOR |
| 22. Dieppe | DIEPPE | | 56. Heysham | HEY |
| 23. Boulogne | BOULOGNE | | 57. Liverpool | LIV |
| 24. Calais | CALAIS | | 58. Liandudno | LLA |
| 25. Dunkerque | DUNKERQUE | | 59. Holyhead | HOL |
| 26. St-Mary's | STM | | 60. Barmouth | BAR |
| 27. Newlyn | NEW | | 61. Fishguard | FIS |
| 28. Devonport | DEV | | 62. Milford Haven | MHA |
| 29. Weymouth | WEY | | 63. Mumbles | MUM |
| 30. Portsmouth | PTM | | 64. New Port | NPO |
| 31. Newhaven | NHA | | 65. Avonmouth | AVO |
| 32. Dover | DOV | | 66. Hinkley Point | HIN |
| 33. Sheerness | SHE | | 67. Ilfracombe | ILF |
| 34. Harwich | HAR | | | |

**Fig. 2.** Location of sites used for the study: 67 ports along the Spanish, French and British coasts. Each site is associated with a number. The table on the right shows the correspondence between numbers and sites. The circled points represent the target sites for which a centered RFA is carried out in this study.

465

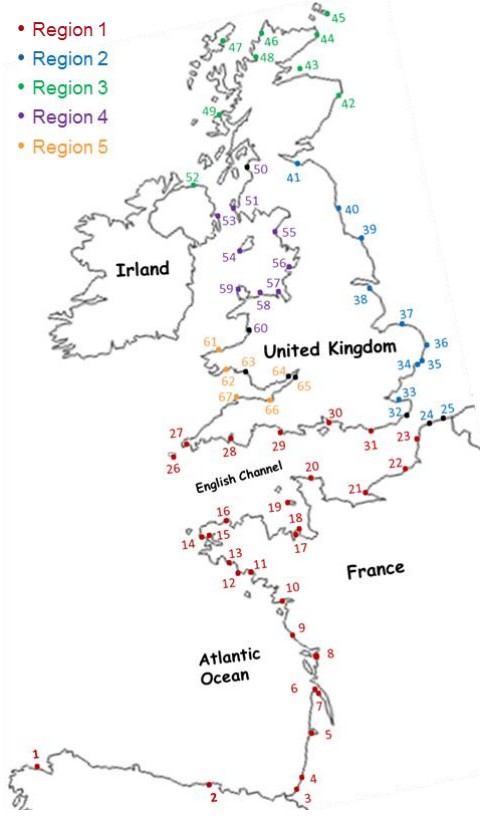

**Fig. 3.** Five physically and statistically homogenous regions (according to Weiss, 2014c). The regions are represented by five colours. This figure shows that, for example, the site 24 (Calais) is located in the region shown in blue and is very close to a border. Site 23 (Boulogne), however very close to site 24 (Calais), is nevertheless in another region (the region shown in red). This separation between site 23 and site 24 may seem artificial.

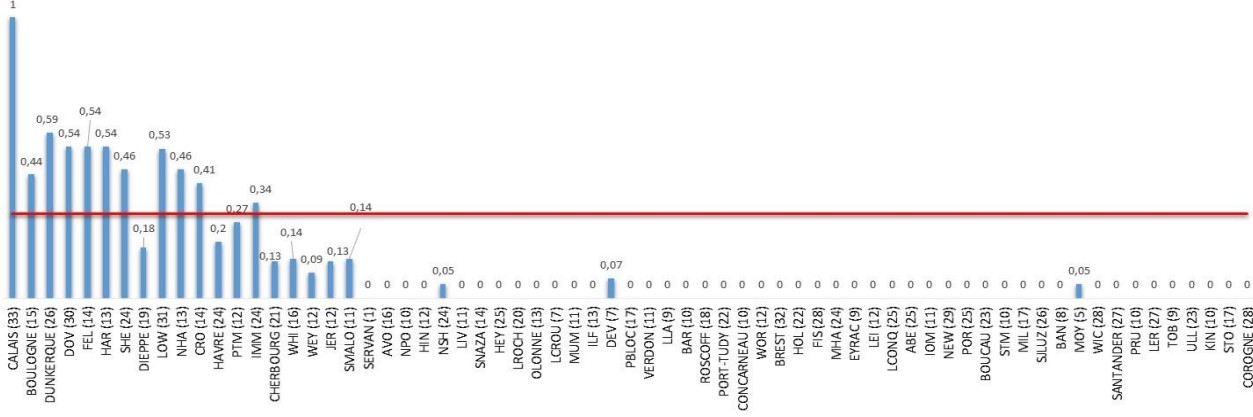

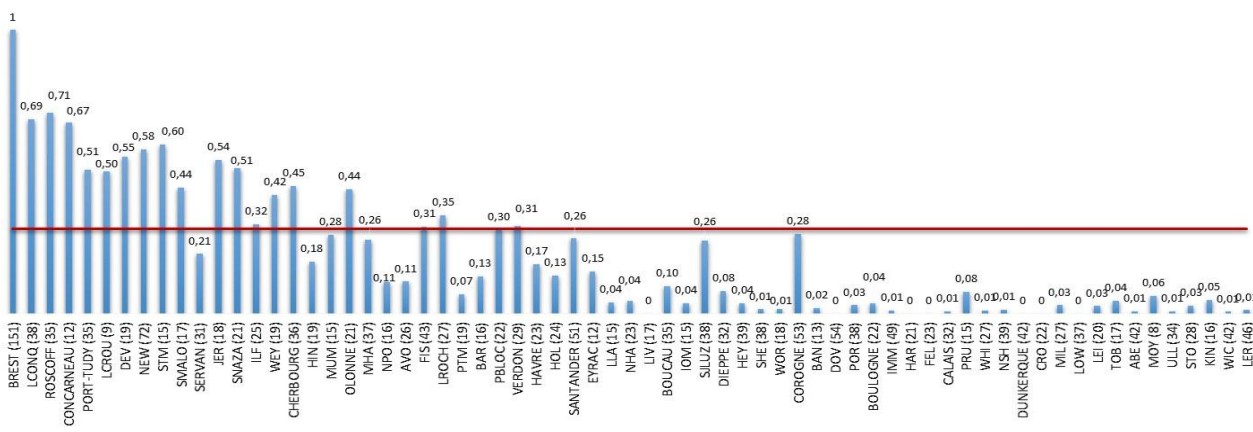

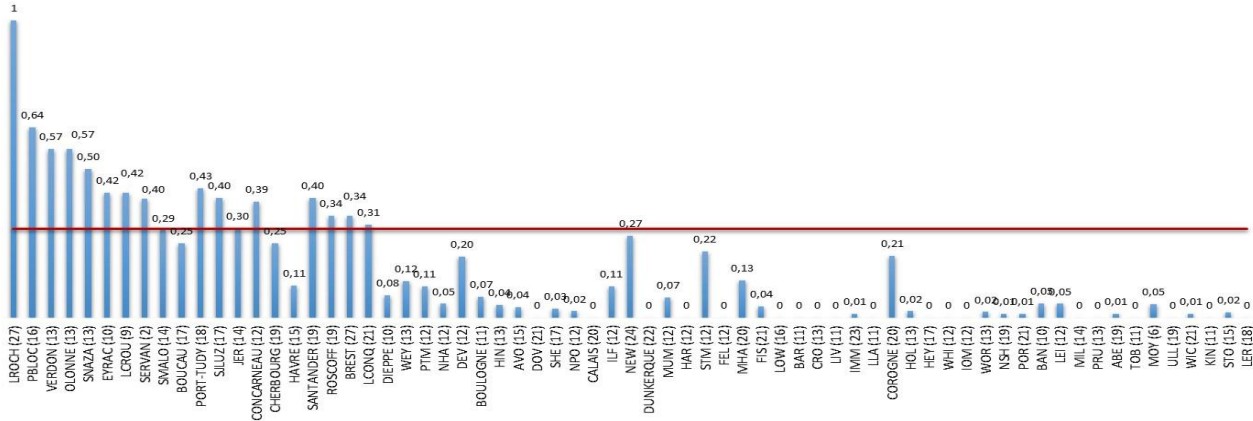

**Fig. 4.** The ESE for Calais (top), Brest (middle) and La Rochelle (bottom). The abbreviations of the horizontal axis are presented in Fig. 2. The vertical blue bars have lengths proportional to the extremogram values. The value of each extremogram is indicated above the blue bars. The red line represents the threshold (equal to 0.3) above which the extremogram value is large enough that the associated site can be considered as belonging to the same region as the target site. The overlap period (in years) of the observation periods of each site with that of the target site is indicated in the brackets next to each site name.

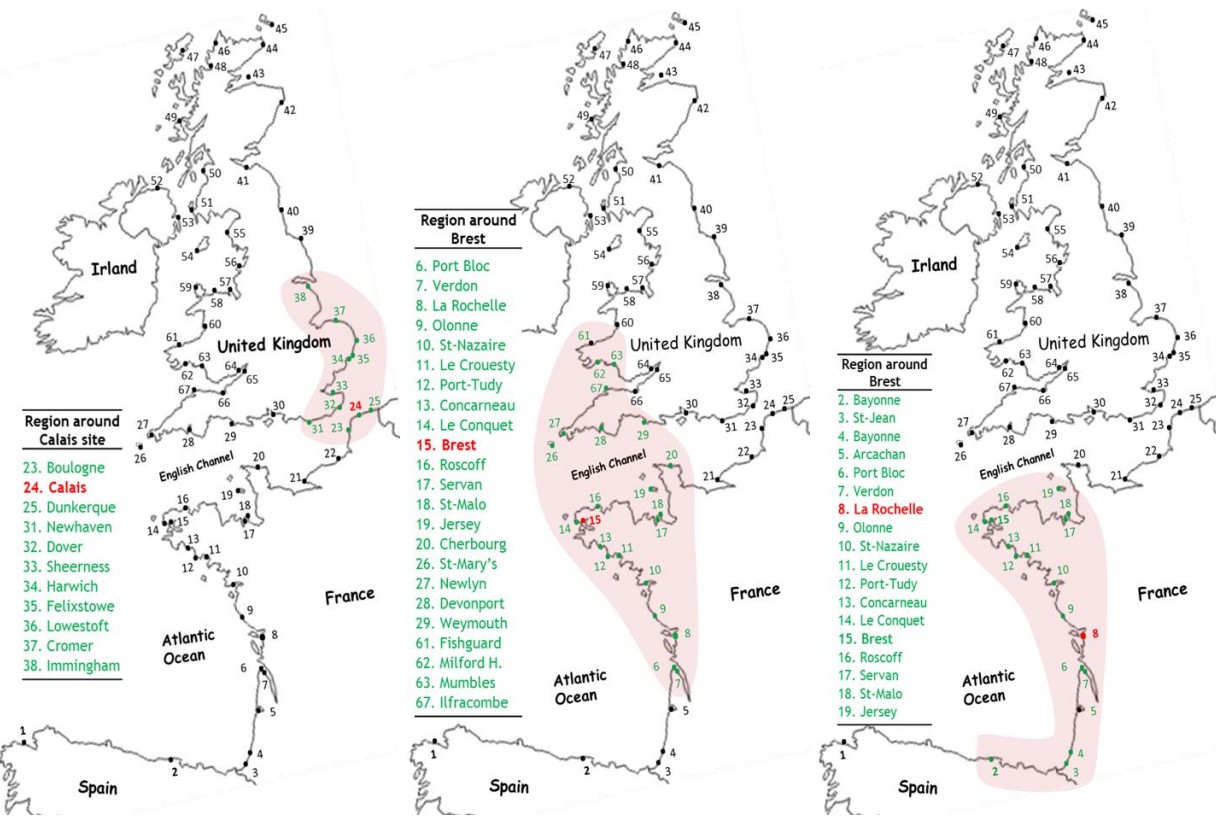

**Fig. 5.** Physically homogenous regions (the list of sites belonging to a region are surrounded by red zones) for Calais, Brest and La Rochelle. Neighbouring sites are represented with green dots (target sites are represented with red dots). Target sites are not close to a border.

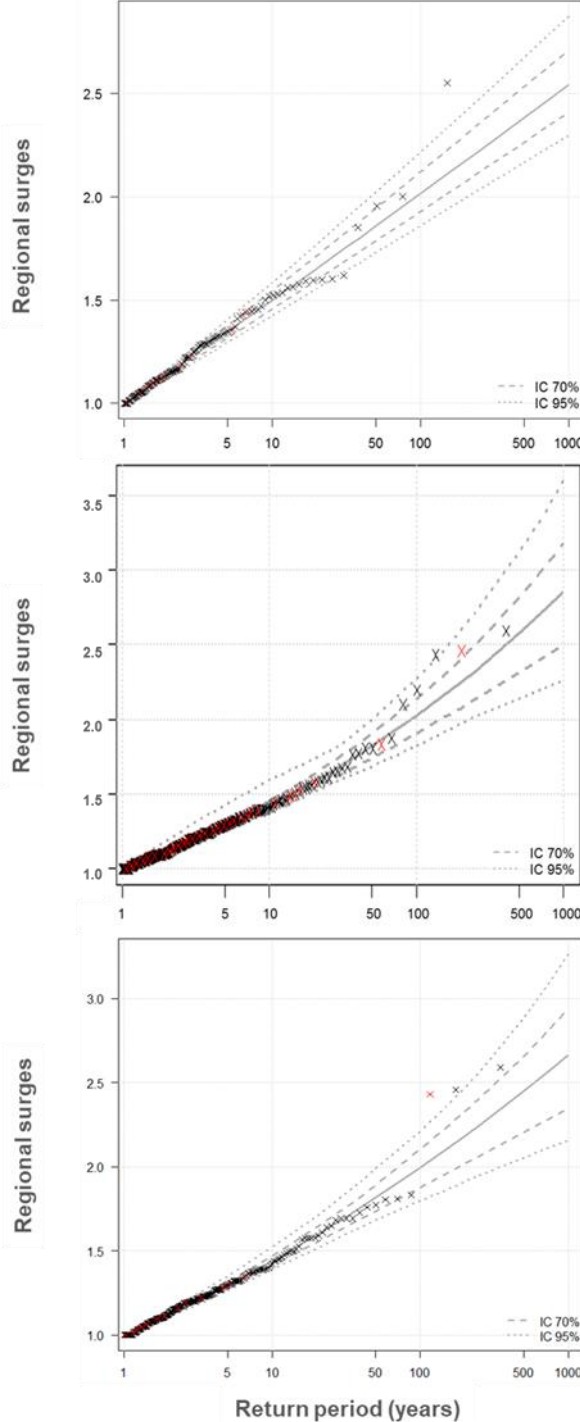

**Fig. 6.** The GPD/sinusoid fitted to regional SSSs (plotting positions, RLs and confidence intervals) for the target sites: $Exp_{sin}$ distribution for Calais (top); $GPD_{cos\,sin}$ for Brest (middle) and La Rochelle (bottom). The y-axis represent the normalized regional surges (without unit), the x-axis represent the return period (in years). The red crosses indicate the SSSs at the target site, so the black crosses indicate the regional ones. The confidence intervals at 70% (dashed line) and 95% (dotted line), which are computed by the bootstrap method, are also presented (dotted lines).

