# Peer review of "Regional frequency analysis of extreme storm surges using the extremogram approach"

_Natural Hazards and Earth System Sciences, 2019_

## Referee Comment (RC1) · Anonymous Referee #1 · 3 Dec 2019

================ General Comments: ================

The present work uses the Empirical Spatial Extremogram (ESE) approach to conduct a Regional Frequency Analysis (RFA) of extreme skew storm surges (SSSs). This is a contribution to our knowledge in flooding hazard in coastal areas.

The authors provide a comprehensive description of the ESE based approach to form a physically homogenous region centred on a target site, avoiding the problem of the "border effect". Once the physically homogeneous regions are formed, the statistical homogeneity is checked and the regional frequency estimation is estimated.

The approach is applied to three target sites (Calais, Brest and La Rochelle) and the results are compared to those obtained in previous work (Weiss, 2014c). The compar-

ison shows progresses concerning on how the statistically homogenous regions are built. The authors should note that the so mentioned work of Weiss (2014c) is missing on the references.

The research falls in the scope of NHESS and the writing is generally good. This article should be interesting to the readership of the journal. This reviewer recommends the article to be published with consideration given to the general and specific comments listed below.

================== Specific Comments: ==================

Line 39: The definition of SSSs is lacking in the introduction.

Line 39: 'Bernardara et al, 2001' not 'Bernardara, Andreewsky and Benoit, 2011'. Note that this also appears in other places in the paper. Please, correct it because when more than two authors are referred in the text the Latin expression 'et al.' needs to be used.

Line 77: 'in section 5.' Not 'in section 5'.

Line 98: the sentence appears with different text size.

Line 147: the definition of lambda, $\lambda$, is lacking.

Line 155: the work of Weiss (2014c) is mentioned several times in the paper but it is missing in the references.

Line 166: The observation period ranges between 1846 and 2011. Why recent years are not considered? What about climate changes in the past? What would happen with projected Sea Level Rise? Is the estimated return period affected? Should this affect the results and its confidence? This should be introduced and discussed in the text.

Line 191: Many times the word neighbours or related words (e.g., neighbouring, neigh-bourhood) are mentioned in the text. Sometimes, it is written 'neighbours' and, other

times, 'neighbors'. Please, make the writing uniform in the paper.

Line 215: Figure 3 is mentioned in the paper. The quality of figure 3 must be improved because the name of the sites are difficult to read. The figure caption should contemplate the meaning of the brackets next to the name of each site.

Line 216: The authors mention Figure 6 instead of Figure 4. Please correct it because this mistake appears several times in the text.

Line 220 and others: 'Dunkirk" or "Dunkerque", "Saint-Malo" or "St-Malo", "Saint-Servan" or "Servan", "Saint Helier" or "Jersey". Note that the sites defined in figures 1 and 3 appear different in the text. Please, avoid it, because this might confuse the readers.

Line 261: the heterogeneity measure H (equal to 0.53) was already ok. Probably the authors want to provide the new value of Dc because that was the discordant parameter. Please, verify it.

Line 280: 'The distribution parameters' not 'The distributions parameters'

Line 289: The paragraph should be rewritten. The definition of RLs is lacking. What are the results in brackets int Table 2? The results and the link to Table 2 values should be better explained because it is confusing.

---

## Referee Comment (RC2) · Anonymous Referee #2 · 17 Feb 2020

**Recommendation**

**Accept with major revisions.** (Mainly textual, presentation)

**Synopsis**

To design coastal protection structures one needs information on the likelihood of extreme water levels, typically in the form of the once-in-N-years event, with N being a large number (100, or 1000, or even higher). Ideally, this information would be obtained from measured time series of a length that is comparable to N. Unfortunately,

such long time series do not exist. The idea of the present paper is to pool time series together from locations that are somehow similar. If several locations are influenced by "the same kind of" surges, the time series from these locations can be combined to obtain a longer series, from which the extremes can be derived more easily. The paper describes the method to decide whether two stations are "similar", and how to combine the time series, and applies it to the western European coastline. As a result three similar regions are detected, one encompassing the whole French coast, a second one centred around the English Channel, and a third one consisting of the eastern (North Sea) coast of England. For these regions, the combination of stations leads to effective lengths of time series of up to 500 years, much longer than the typical 50 years of individual stations.

**Discussion**

The paper addresses a very important topic, namely the estimation of rare events. Pooling measurements from different locations makes it possible to extract more information from the existing short times series and allows to obtain a sharper estimate of the once-in-N-years event than it is possible from individual series. The paper is based on a sound mathematical concept. Therefore, it should be published. However, I found it hard to follow. Therefore, I recommend a partial re-writing to make the paper easier to understand for readers not fully familiar with the topic. Below I give some more detailed suggestions.

**Detailed comments**

**p 2, l 51** You doubt that Boulonge-Sur-Mer and Calais belong to two different regions because they are so close together. However, is mere distance a good argument?

The two locations face different seas. Calais is oriented to the North Sea, while Boulonge faces the Channel and perhaps even the Atlantic. So I do not see an a priory reason why they could not belong to different regions in terms of surge heights.

**eq. (1)** How are the intervals $A$ and $B$ defined?

**eq. (2)** Comparing with (1), shouldn't it read $X(t+h) > q_X$, and $Y(t) > q_Y$?

**eq. (2)** Why is the upper bound of the summation in the nominator given by $D - h$? According to the explanation below the equation, $D$ is the number of events, while $h$ is a measure of time. $D - h$ then does not make much sense to me. In the denominator, the upper bound is $N$. How is $N$ defined, and why are the upper bounds of summation different in nominator and denominator?

**p 2, l 101** I guess $I_f = I\{f\}$ to conform with the notation in eq. (2). Please clarify.

**p 2, l 101** $D$ is the number of events that "occurred at the same time" - I think it should be "that occurred within a time of $h$" at the two sites.

**p 2, l 103** Why does the fact that $\hat{\rho} \in (0,1]$ indicate a dependency between $X$ and $Y$? What would $\hat{\rho}$ be in case of no dependency?

**together** Please explain the the equations in more detail and make sure that the notation is consistent.

**p 2, l 104** Start a new paragraph before "Let $\rho_0$ be ..."

**p 3, l 108** What do you mean by "confusion"?

**p 3, l 133-138** Please give the definitions of $H$ and $D_c$, perhaps in an appendix.

**p 4, l 141** "duration" - I guess you mean "length of time series"? I was confused because I associated duration with "duration of a storm". This applies for the whole paragraph.

**sect. 2.3** General: As far as I understand there are two issues: First, to define "Neighbourhood", you need time series that overlap as much as possible. Second, to obtain as many independent events as possible, you would prefer non-overlapping time series. The effective length of overlapping series is shorter than their sum because some events are "extreme" in both locations and thus not independent. You should clarify these concepts and perhaps show a figure as an example (appendix?).

**p 4, l 147** You introduce $\lambda_r$, but the explanation is given only after the equation. This is confusing. Define a symbol when you introduce it. But more important: What is $\varphi$? If you say that it was shown that $\varphi = \lambda_r/\lambda$, then you have to say first what $\varphi$ stands for.

**eq. (3)** Try to interpret this equation. I cannot recover the results for the limiting cases "completely (in)dependent".

**p 5, l 162** predicted tide - you mean astronomical tide? And please give a definition of SSS. Is it water level, residual surge, or skew surge, or something else?

**p 5, l 167** How is "greatest majority" defined? Serious: I think "majority" suffices.

**sect. 3.1** Again I would like to see a figure with an example.

**p 7, l 211** Throughout the paper you refer a lot to Weiss (2014c), sometimes saying "confirming Weiss", sometime saying "different from Weiss". I have the impression (but I may be wrong) that one of the goals of the paper is to improve upon

the results of Weiss. If so, you should clearly state this at the beginning of the paper and explain how Weiss did his analysis, what he found, and what the present paper is improving.

**p 7, l 221**  duration → length of record?

**p 7/8, l 210-254**  Figure 6 → Figure 4 (everywhere)

**p 8, l 258**  add the station number (5) to the name. That makes it easier to fin the station on the maps (also for other stations when they are mentioned)

**p 8, l 271**  imply → implies

**p 8, l 272**  duration → length (?) (2×)

**p 8 l, 275**  show → shown

**p 9, l 280**  law → distribution

**p 9, l 282/283**  Define/explain $Exp_{sin}$ and $Gpd_{cos\,sin}$.

**p 9, l 292**  Formulation: *The same conclusion is not the same* - sounds strange

**p 9, l 300**  Why *interesting*?

**p 9, l 304**  Again, which finding is interesting, and why?

**p 9, l 306-308**  See my remark about referring to Weiss above. - Why is the finding that that regions 1 and 2 are statistically homogeneous, a progress?

**p 10, l 312**  consider → considered

**p 10, l 313**  take → taken; remove → removing

**p 10, l 313**  removing a site - the criterion should be the $D_c$-value, shouldn't it?

**p 10, l 314/315** please show Dieppe - otherwise the remark cannot be understood

**p 10, l 315** *not centered enough on the target site* - what do you mean?

**p 10, ll 326** duration → length

**Fig. 2** Why do you reproduce a figure from Weiss? See my remark above.

**Fig. 3** The black background makes the figure hard to read. Try a background that gives a higher contrast to the other elements (bars, red line, numbers) in the plot.

**Fig. 3** Give value for red line - $\sim 0.3$, I guess.

**Fig. 5** Units for the y-axis (Regional surge)?

**Fig. 5, caption** GPDcos sin → Gpd$_{\text{cos sin}}$; similarly for Expsin

**Fig. 5, caption** 70% = dashed line, 95% = dotted line.
* * *

---

## Author Comment (AC1) · 28 Mar 2020

Dear Referee #1,

Thank you so much for reviewing our paper.

The manuscript will be, therefore, modified to consider your constructive comments. In the following, a point-by-point response to your comments will be presented.

**Specific comments:**

| line | comment | response |
|------|---------|----------|
| 39 | The definition of SSSs is lacking in the introduction. | We agree with referee #1. The following sentence was added in §2 – section 1 (introduction): lines 42-44 "For convenience, we would like to recall here the definition of a skew surge: it is the difference between the maximum observed water level and the maximum predicted tidal level regardless of their timing during the tidal cycle (a tidal cycle contains one skew surge)." |
| 39 | 'Bernardara et al, 2001' not 'Bernardara, Andreewsky and Benoit, 2011'. Note that this also appears in other places in the paper. Please, correct it because when more than two authors are referred in the text the Latin expression 'et al.' needs to be used. | Corrected. |
| 77 | 'in section 5.' Not 'in section 5'. | OK. |
| 98 | the sentence appears with different text size. | Fixed. |
| 147 | the definition of lambda, $\lambda$, is lacking. | We agree with referee #1. The following sentence was added to the text (line 171): "$\lambda$ is the average number of storms per year at each site." |
| 155 | the work of Weiss (2014c) is mentioned several times in the paper but it is missing in the references. | That's right. The reference was added to the references list. |
| 166 | 1- The observation period ranges between 1846 and 2011. Why recent years are not considered?

2- What about climate changes in the past? What would happen with projected Sea Level Rise? Is the estimated return period affected? Should this affect the results and its confidence? This should be introduced and discussed in the text. | 1- the following sentence was added at the beginning of §2 section 2.5 (line 204): "For convenience, the same observation periods as those used by Weiss (2014c) have been used in the present study. "

2- the following sentence was added at the end of §2 section 2.5

"It is also noteworthy that the impact of climate change on the estimated return levels and associated uncertainties is not covered by this paper. The use of projected sea level rise could however be the object of another paper." |
| 191 | Many times the word neighbours or related words (e.g., neighbouring, neighbourhood) are mentioned in the text. Sometimes, it is written 'neighbours' and, other times, 'neighbors'. Please, make the writing uniform in the paper. | We agree with referee #1. Writing made uniform. |

| | | |
|---|---|---|
| 215 | Figure 3 is mentioned in the paper. The quality of figure 3 must be improved because the name of the sites are difficult to read. The figure caption should contemplate the meaning of the brackets next to the name of each site. | Ok. Quality of figure 3 improved. |
| 216 | The authors mention Figure 6 instead of Figure 4. Please correct it because this mistake appears several times in the text. | Done. |
| 220 and others: | 'Dunkirk" or "Dunkerque", "Saint-Malo" or "St-Malo", "Saint- Servan" or "Servan", "Saint Helier" or "Jersey". Note that the sites defined in figures 1 and 3 appear different in the text. Please, avoid it, because this might confuse the readers. | Done. |
| 261 | the heterogeneity measure H (equal to 0.53) was already ok. Probably the authors want to provide the new value of Dc because that was the discordant parameter. Please, verify it. | We agree with referee #1. Verified. |
| 280 | 'The distribution parameters' not 'The distributions parameters' | Done. |
| 289 | 1- The paragraph should be rewritten.
2- The definition of RLs is lacking.

3- What are the results in brackets int Table 2? The results and the link to Table 2 values should
be better explained because it is confusing. | 1- The § was rewritten.

2- Return level defined. Please see 2$^{nd}$ sentence - §1 – section 1 (Introduction): lines 27-28: "The $T$-year return level can be defined as a high quantile for which the probability that an extreme value (the annual maximum, for instance) exceeds this quantile is $\frac{1}{T}$ ."

3- Results in brackets are confidence interval width (fixed in the title of table 2) |
| | | |

---

## Author Comment (AC2) · 28 Mar 2020

Dear Referee #2,

Thank you so much for reviewing our paper.

The manuscript will be, therefore, modified to consider your constructive comments. In the following, a point-by-point response to your comments will be presented.

**Detailed comments:**

| page | Line/equation | comment | response |
|---|---|---|---|
| 2 | 51 | 1- You doubt that Boulonge-Sur-Mer and Calais belong to two different regions because they are so close together. However, is mere distance a good argument?

2- The two locations face different seas. Calais is oriented to the North Sea, while Boulonge faces the Channel and perhaps even the Atlantic. So I do not see an a priory reason why they could not belong to different regions in terms of surge heights. | We agree with referee #2. This is a very interesting comment.

Indeed, Calais is located on the Opal Coast, at the edge of the Pas de Calais which is at the limit between the English Channel and the North Sea, facing the English coast, while Boulogne is located on the edge of the English Channel, also facing the English coast.
The climate comparison (url provided) from the website of Meteo France indicates that those 2 cities have the same climate.

the text has been changed to take it into account (lines 54-58)

" For instance, in the regions of interest obtained by Weiss et al. (2013), the two French sites, Boulogne-Sur-Mer and Calais are located in two different regions while they are geographically close with a distance of about 30 km. Indeed, despite the fact that the two sites face different seas (North sea and English chanel), they have the same climate (according to the Meteo-France's climate comparator). " |
| 3 | eq. (1)

eq.(2)

eq. (2) | - How are the intervals A and B defined?
- Comparing with (1), shouldn't it read $X(t+h) > q_X$ and $Y(t) > q_Y$ ?
- Why is the upper bound of the summation in the nominator given by $D-h$ ?
According to the explanation below the equation, $D$ is the number of events, while $h$ is a measure of time. $D-h$ then does not make much sense to me. In the denominator, the upper bound is $N$. How is $N$ defined, and why are the upper bounds of summation different in nominator and denominator? | We agree with referee #2 A more clearer and straightforward explanation of the empirical extremogram is rather proposed. Intervals A and B and time lag h no longer appear in the equations.

We agree with referee #2.
$D$ (Instead of $D-h$ ) and $N$ are used now. |
| 2 | 101 | I guess $I_f = I\{f\}$ to conform with the notation in eq. (2). Please clarify. | Yes. Clarified. |

| 2 | 101 | $D$ is the number of events that "occurred at the same time" - I think it should be "that occurred within a time of $h$" at the two sites. | That was right. But there is no longer time lag $h$ in the equation |
|---|---|---|---|
| 2 | 103 | Why does the fact that $\rho \in (0,1]$ indicate a dependency between X and Y ?

 What would $\rho$ be in case of no dependency? | 0 is a very small dependency and 1 is a very high dependency (site with itself).

 2- In case of no dependency, $\rho$ would be equal to 0. |
| 3 | eqs(1,2) | Please explain the the equations in more detail and make sure that the notation is consistent. | We agree with referee #2. The equations were modified and are clearer now. |
| 2 | 104 | Start a new paragraph before "Let $\rho_0$ be ..." | Done. |
| 3 | 108 | What do you mean by "confusion"? | Good point. The sentence is deleted. |
| 3 | 133-138 | Please give the definitions of $H$ and $D_c$ , perhaps in an appendix | Line 155: $H$ is the heterogeneity indicator which is a measure of whether the dispersion between sites.

 Lines 156-158: The discordancy criterion $D_c$ to ensure that any site is not significantly different, in terms of L-moments, from the other sites. Hosking and Wallis suggest that a site is discordant if $D_c < 3$ . |
| 4 | 141 | "duration" - I guess you mean "length of time series"? I was confused because I associated duration with "duration of a storm". This applies for the whole paragraph. | Yes it is. The term duration of observations is used instead. |
| sect. 2.3 – General | | As far as I understand there are two issues: First, to define "Neighbourhood", you need time series that overlap as much as possible. Second, to obtain as many independent events as possible, you would prefer nonoverlapping time series. The effective length of overlapping series is shorter than their sum because some events are "extreme" in both locations and thus not independent. You should clarify these concepts and perhaps show a figure as an example (appendix?). | Very interesting comment.

 This may seem paradoxical. So, on the one hand we use the dependency (when we need it), on the other hand we bypass it.

 And it works! |
| 4 | 147 | 1- Define a symbol when you introduce it
 2- But more important: What is $\varphi$ ?
 If you say that it was shown that $\varphi = \lambda_r / \lambda$, then you have to say first what $\varphi$ stands for. | 1- done

 2- more explanation about $\varphi$ is proposed. |

| | | | |
|---|---|---|---|
| | eq(3) | Try to interpret this equation. I cannot recover the results for the limiting cases "completely (in)dependent". | We agree with referee #2. Equation 3 is better interpreted in section 2.3. The two limiting cases are then clearly recovered. |
| 5 | 162 | predicted tide - you mean astronomical tide? And please give a definition of SSS. Is it water level, residual surge, or skew surge, or something else? | Done. |
| 5 | 167 | How is "greatest majority" defined? Serious: I think "majority" suffices. | Right. |
| sect. 3.1 | | Again I would like to see a figure with an example. | A figure was added (fig. 1): An illustrative example on how the extremal dependency coefficient is empirically computed. |
| 7 | 211 | Throughout the paper you refer a lot to Weiss (2014c), sometimes saying "confirming Weiss", sometime saying "different from Weiss". I have the impression (but I may be wrong) that one of the goals of the paper is to improve upon the results of Weiss. If so, you should clearly state this at the beginning of the paper and explain how Weiss did his analysis, what he found, and what the present paper is improving. | Even Weiss' work is interesting, we still try to improve the methodology of constructing the homogeneous region (to be used in a regional frequency analysis). We added the following sentence in the introduction section (begining of §3) to clearly state that: "To address this limitation (the border-effect problem) and to form a physically homogeneous region centered on a target site, ..." |
| 7 | 221 | Duration → length of record | OK. The term duration of observations is used instead. |
| 7-8 | 210-254 | Figure 6 → Figure 4 (everywhere) | OK. |
| 8 | 258 | add the station number (5) to the name. That makes it easier to fin the station on the maps (also for other stations when they are mentioned) | Done. |
| 8 | 271 | Imply → implies | OK. |
| 8 | 272 | Duration → length (?) | OK. The term duration of observations is used instead. |
| 8 | 275 | Show → shown | OK. |
| 9 | 280 | Law → distribution | OK. |
| 9 | 282-283 | Define/explain $Exp_{sin}$ and $Gpd_{cos}$ sin. | the reader is referred to Weiss (2014c) for definition/explanation of these frequency models |
| 9 | 292 | Formulation: The same conclusion is not the same - sounds strange | OK. |
| 9 | 300 | Why interesting? | replaced by: " ... to a certain geographical consistency" |
| 9 | 304 | Again, which finding is interesting, and why? | particularly interesting for us because:
- they are no longer close to the limit of a region and
- since they can be representative sites for the Gravelines nuclear power plant in France. |

| | | | added in the text |
|---|---|---|---|
| 9 | 306-308 | See my remark about referring to Weiss above. - Why is the finding that regions 1 and 2 are statistically homogeneous, a progress? | What is new here is that the regions of interest are statistically homogeneous and centered on target sites.

Made clearer in the paper. |
| 10 | 312 | Consider → considered | OK. |
| 10 | 313 | Take → taken;
remove → removing | OK. |
| 10 | 313 | removing a site - the criterion should be the Dc-value, shouldn't it? | Yes, as well. you are right. |
| 10 | 314-315 | please show Dieppe - otherwise the remark cannot be understood | Sentence deleted. |
| 10 | 315 | not centered enough on the target site - what do you mean? | the sentence was deleted. |
| 10 | 326 | Duration → length | OK. The term duration of observations is used instead. |
| Fig. 2 | | Why do you reproduce a figure from Weiss? See my remark above. | We reproduce the results obtained by Weiss because it makes things easier to show the border effect in his results and it is easier for us explain things with a supporting graphic. |
| Fig. 3 | | The black background makes the figure hard to read. Try a background that gives a higher contrast to the other elements (bars, red line, numbers) in the plot. | OK. |
| Fig. 3 | | Give value for red line - 0:3, I guess. | OK. |
| Fig. 3 | | Units for the y-axis (Regional surge)? | OK. |
| Fig. 3 | | GPDcos $_{sin}$ → Gpdcos sin; similarly for Exp$_{sin}$ | OK. |
| Fig. 3 | | 70% = dashed line, 95% = dotted line. | OK. |

---

## Author Response (AR1)

Dear Referees,

We would like to thank you again for reviewing our paper and for the constructive comments.

You will find 2 additional attached files:

The Word document (contained in the zip archive) contains the list of the corrections made: both the old and the new version are visible in the same document.
The PDF file contains the latest version (which is not in correction mode).

Your comments are constructive and helped us to improve our paper. The point-by-point responses to all your comments are, moreover, contained in other files already uploaded

Yours sincerely,
Marc Andreevsky and Yasser Hamdi on behalf of the co-authors

[revised manuscript text omitted]

---

## Referee Report (RR1)

Review of

**Use of empirical spatial extremogram to define homogeneous regions for a regional frequency analysis of extreme storm surges**

by M. Andreevsky, Y. Hamdi, S. Griolet, P. Bernardara, and R. Frau

- Revised Version -

**Recommendation**

**Accept with minor revisions.** (Mainly textual, presentation)

**Discussion**

The authors have adequatly addressed my suggestions and questions I had on their original submission. The mathematical explanation of the ESE is much clearer now. Nevertheless, I have some remarks and suggestions for improvement.

**Detailed comments**

The line numbers refer to the document *nhess-2019-277-manuscript-version3.pdf*.

**p 3/4, l 90-115** The notation could be made more consistent. Sometimes, the sites are denoted by subscripts "1" and "2" (e.g., eq. 1 and preceding text), and sometimes as "X" and "Y" (eq. 2 and *parts* of following text).

**same para** I miss the notion of 'simultaneity'. If you consider the probability that $P[X \geq q_1 \mid Y \geq q_2]$ (eq. 1), then you have to say something about the time difference between the events at the two sites that you allow for to consider them as 'same'.

**p 5, l 166** Is this the definition of $\varphi$? Then the sentence should not start with *It was shown that $\varphi =$*. That beginning of a sentence implies that $\varphi$ has been introduced before, and that it is now stated that it can be expressed as $\lambda_r/\lambda$. But I cannot find a definition of $\varphi$ anywhere else. Furthermore, the symbol $\varphi$ is not used anywhere else in the paper. So what is the use of defining it?

**p 5, l 209/210** This definition of SSS is usually called the *skew surge*.

---

## Author Response (AR2)

Dear Referee #2,

Thank you for reviewing our paper. The manuscript will be, therefore, modified to consider your comments. In the following, a point-by-point response to your comments will be presented.

**Detailed comments:**

| Page/line | Comment | Response |
|---|---|---|
| p 3/4, l 90-115 | The notation could be made more consistent. Sometimes, the sites are denoted by subscripts "1" and "2" (e.g., eq. 1 and preceding text), and sometimes as "X" and "Y" (eq. 2 and parts of following text). | It is corrected |
| p 3/4, l 90-115 | Same para I miss the notion of 'simultaneity'. If you consider the probability that $P[X < q_1 \ \& \ Y < q_2]$ (eq. 1), then you have to say something about the time difference between the events at the two sites that you allow for to consider them as 'same'. | The word 'simultaneity' is suppressed. We add instead: "… extreme events generated by the same storm that occurred at sites S1 and S2. It was considered that two observations were generated by the same storm if the time difference between their two moments of occurrence was less, in absolute value, than 6 hours (other durations were tested, 12h and 24h but the results were similar). " |
| p 5, l 166 | Is this the definition of $\varphi$? Then the sentence should not start with It was shown that $\varphi =$. That beginning of a sentence implies that ' has been introduced before, and that it is now stated that it can be expressed as $\lambda_r = \lambda$. But I cannot find a definition of $\varphi$ anywhere else. Furthermore, the symbol $\varphi$ is not used anywhere else in the paper. So what is the use of defining it? | It is corrected. $\varphi$ comes from Weiss et all. (2014b). We have added this precision. |
| p 5, l 209/210 | This definition of SSS is usually called the skew surge. | We used the same notation than the following publication in NHESS "Modelling dependence and coincidence of storm surges and high tide: |

| | | Methodology and simplified case study in Le Havre (France) " (Amine Ben Daoued et all., 2019, NHESS) which uses both SS and SSS notations. There are 3 words (Skew Storm Surge), so we found it appropriate to use 3 letters. This acronym is already used in another NHESS publication but if it is really annoying, we can change it. |
| --- | --- | --- |